# Direct Feedback Alignment Scales to Modern Deep Learning Tasks and Architectures

**Julien Launay**[1,2]    **Iacopo Poli**[1]    **François Boniface**[1]    **Florent Krzakala**[1,2,3]

[1]LightOn    [2]LPENS, École Normale Supérieure    [3] IdePhics, EPFL

{julien, iacopo, francois, florent}@lighton.ai

## Abstract

Despite being the workhorse of deep learning, the backpropagation algorithm is no panacea. It enforces sequential layer updates, thus preventing efficient parallelization of the training process. Furthermore, its biological plausibility is being challenged. Alternative schemes have been devised; yet, under the constraint of synaptic asymmetry, none have scaled to modern deep learning tasks and architectures. Here, we challenge this perspective, and study the applicability of Direct Feedback Alignment (DFA) to neural view synthesis, recommender systems, geometric learning, and natural language processing. In contrast with previous studies limited to computer vision tasks, our findings show that it successfully trains a large range of state-of-the-art deep learning architectures, with performance close to fine-tuned backpropagation. When a larger gap between DFA and backpropagation exists, like in Transformers, we attribute this to a need to rethink common practices for large and complex architectures. At variance with common beliefs, our work supports that challenging tasks can be tackled in the absence of weight transport.

## 1 Introduction

While the backpropagation algorithm (BP) [1, 2] is at the heart of modern deep learning achievements, it is not without pitfalls. For one, its weight updates are non-local and rely on upstream layers. Thus, they cannot be easily parallelized [3], incurring important memory and compute costs. Moreover, its biological implementation is problematic [4, 5]. For instance, BP relies on the transpose of the weights to evaluate updates. Hence, synaptic symmetry is required between the forward and backward path: this is implausible in biological brains, and known as the weight transport problem [6].

Consequently, alternative training algorithms have been developed. Some of these algorithms are explicitly biologically inspired [7–13], while others focus on making better use of available compute resources [3, 14–19]. Despite these enticing characteristics, none has been widely adopted, as they are often demonstrated on a limited set of tasks. Moreover, as assessed in [20], their performance on challenging datasets under the constraint of synaptic asymmetry is disappointing.

We seek to broaden this perspective, and demonstrate the applicability of Direct Feedback Alignment (DFA) [19] in state-of-the-art settings: from applications of fully connected networks such as neural view synthesis and recommender systems, to geometric learning with graph convolutions, and natural language processing with Transformers. Our results define new standards for learning without weight transport and show that challenging tasks can indeed be tackled under synaptic asymmetry.

All code is available on the paper website at `lair.lighton.ai/dfa-scales`.

## 1.1 Related work

Training a neural network is a credit assignment problem: an update is derived for each parameter from its contribution to a cost function. To solve this problem, a spectrum of algorithms exists [21].

**Biologically motivated methods**   Finding a training method applicable under the constraints of biological brains remains an open problem. End-to-end propagation of gradients is unlikely to occur [22], implying local learning is required. Furthermore, the weight transport problem enforces synaptic asymmetry [6]. Inspired by auto-encoders, target propagation methods (TP) [10–12] train distinct feedback connections to invert the feedforward ones. Feedback alignment (FA) [13] replaces the transpose of the forward weights used in the backward pass by a random matrix. Throughout training, the forward weights learn to *align* with the arbitrary backward weights, eventually approximating BP.

**Beyond biological considerations**   As deep learning models grow bigger, large-scale distributed training is increasingly desirable. Greedy layer-wise training [14] allows networks to be built layer by layer, limiting the depth of backpropagation. To enable parallelization of the backward pass, updates must only depend on local quantities. Unsupervised learning is naturally suited for this, as it relies on local losses such as Deep InfoMax [17] and Greedy InfoMax [18]. More broadly, synthetic gradient methods, like decoupled neural interfaces [3, 15] and local error signals (LES) [16], approximate gradients using layer-wise trainable feedback networks, or using reinforcement learning [23]. DFA [19] expands on FA and directly projects a global error to each layer. A shared feedback path is still needed, but it only depends on a simple random projection operation.

**Performance of alternative methods**   Local training methods are successful in unsupervised learning [18]. Even in a supervised setting, they scale to challenging datasets like CIFAR-100 or ImageNet [14, 16]. Thus, locality is not too penalizing. However, FA, and DFA are unable to scale to these tasks [20]. In fact, DFA is unable to train convolutional layers [24], and has to rely on transfer learning in image tasks [25]. To enable feedback alignment techniques to perform well on challenging datasets, some form of weight transport is necessary: either by explicitly sharing sign information [26–28], or by introducing dedicated phases of alignment for the forward and backward weights where some information is shared [29, 30]. To the best of our knowledge, no method compatible with the weight transport problem has ever been demonstrated on challenging tasks.

## 1.2 Motivations and contributions

We focus on DFA, a compromise between biological and computational considerations. Notably, DFA is compatible with synaptic asymmetry: this asymmetry raises important challenges, seemingly preventing learning in demanding settings. Moreover, it allows for asynchronous weight updates, and puts a single operation at the center of the training stage. This enables new classes of training co-processors [31, 32], leveraging dedicated hardware to perform the random projection.

**Extensive survey**   We apply DFA in a large variety of settings matching current trends in machine learning. Previous works have found that DFA is unsuitable for computer vision tasks [20, 24]; but computer vision alone cannot be the litmus test of a training method. Instead, we consider four vastly different domains, across eight tasks, and with eleven different architectures. This constitutes a survey of unprecedented scale for an alternative training method, and makes a strong case for the possibility of learning without weight transport in demanding scenarios.

**Challenging settings**   We demonstrate the ability of DFA to tackle challenging tasks. We successfully learn and render real-world 3D scenes (section 3.1.1); we perform recommendation at scale (section 3.1.2); we explore graph-based citation networks (section 3.2); and we consider language modelling with a Transformer (section 3.3). We study tasks at the state-of-the-art level, that have only been recently successfully tackled with deep learning.

**Modern architectures**   We prove that the previously established failure of DFA to train convolutions does not generalize. By evaluating performance metrics, comparing against a shallow baseline, measuring alignment, and visualizing t-SNE embeddings, we show that learning indeed occurs in layers involving graph convolutions and attention. This significantly broadens the applicability of DFA–previously thought to be limited to simple problems like MNIST and CIFAR-10.

## 2 Methods

**Forward pass**   In a fully connected network, at layer $i$ out of $N$, neglecting its biases, with $\mathbf{W}_i$ its weight matrix, $f_i$ its non-linearity, and $\mathbf{h}_i$ its activations, the forward pass is:

$$\forall i \in [1, \ldots, N] : \mathbf{a}_i = \mathbf{W}_i \mathbf{h}_{i-1}, \mathbf{h}_i = f_i(\mathbf{a}_i). \tag{1}$$

$\mathbf{h}_0 = X$ is the input data, and $\mathbf{h}_N = f(\mathbf{a}_N) = \hat{\mathbf{y}}$ are the predictions. A task-specific cost function $\mathcal{L}(\hat{\mathbf{y}}, \mathbf{y})$ is computed to quantify the quality of the predictions with respect to the targets $\mathbf{y}$.

**Backward pass with BP**   The weight updates are computed by backpropagation of the error vector. Using the chain-rule of derivatives, each neuron is updated based on its contribution to the cost function. Leaving aside the specifics of the optimizer used, the equation for the weight updates is:

$$\delta \mathbf{W}_i = -\frac{\partial \mathcal{L}}{\partial \mathbf{W}_i} = -[(\mathbf{W}_{i+1}^T \delta \mathbf{a}_{i+1}) \odot f_i'(\mathbf{a}_i)]\mathbf{h}_{i-1}^T, \delta \mathbf{a}_i = \frac{\partial \mathcal{L}}{\partial \mathbf{a}_i} \tag{2}$$

**Backward pass with DFA**   The gradient signal $\mathbf{W}_{i+1}^T \delta \mathbf{a}_{i+1}$ of the (i+1)-th layer violates synaptic asymmetry. DFA replaces it with a random projection of the topmost derivative of the loss, $\delta \mathbf{a}_y$. For common classification and regression losses such as the mean squared error or the negative log likelihood, this corresponds to a random projection of the global error $\mathbf{e} = \hat{\mathbf{y}} - \mathbf{y}$. With $B_i$, a fixed random matrix of appropriate shape drawn at initialization for each layers:

$$\delta \mathbf{W}_i = -[(\mathbf{B}_i \delta \mathbf{a}_y) \odot f_i'(\mathbf{a}_i)]\mathbf{h}_{i-1}^T, \delta \mathbf{a}_y = \frac{\partial \mathcal{L}}{\partial \mathbf{a}_y} \tag{3}$$

We provide details in appendix C regarding adapting DFA beyond fully-connected layers.

## 3 Experiments

We study the applicability of DFA to a diverse set of applications requiring state-of-the-art architectures. We start with fully connected networks, where DFA has already been demonstrated, and address new challenging settings. We then investigate geometric learning: we apply DFA to graph neural networks in classification tasks on citation networks, as well as graph autoencoders. These architectures feature graph convolutions and attention layers. Finally, we use DFA to train a transformer-based Natural Language Processing (NLP) model on a dataset of more than 100 million tokens.

### 3.1 Fully connected architectures

DFA has been successful at training fully connected architectures, with performance on-par with backpropagation [19, 20]. However, only computer vision tasks have been considered, where fully connected networks considerably underperform their convolutional counterpart. Here, we focus on tasks where fully connected architectures are state-of-the-art. Moreover, the architectures considered are deeper and more complex than those necessary to solve a simple task like MNIST.

#### 3.1.1 Neural view synthesis with Neural Radiance Fields

The most recent state-of-the-art *neural view synthesis* methods are based on large fully connected networks: this is an ideal setting for a first evaluation of DFA on a challenging task.

**Background**   There has been growing interest in methods capable of synthesising novel renders of a 3D scene using a dataset of past renders. The network is trained to learn an inner representation of the scene, and a classical rendering system can then query the model to generate novel views. With robust enough methods, real-world scenes can also be learned from a set of pictures.

Until recently, most successful neural view synthesis methods were based on sampled volumetric representations [33–35]. In this context, Convolutional Neural Networks (CNNs) can be used to smooth out the discrete sampling of 3D space [36, 37]. However, these methods scale poorly to higher resolutions, as they still require finer and finer sampling. Conversely, alternative schemes based on a continuous volume representation have succeeded in generating high-quality renders [38], even featuring complex phenomenons such as view-dependant scattering [39]. These schemes make point-wise predictions, and use fully connected neural networks to encode the scene. Beyond 3D scenes, continuous implicit neural representations can be used to encode audio and images [40].

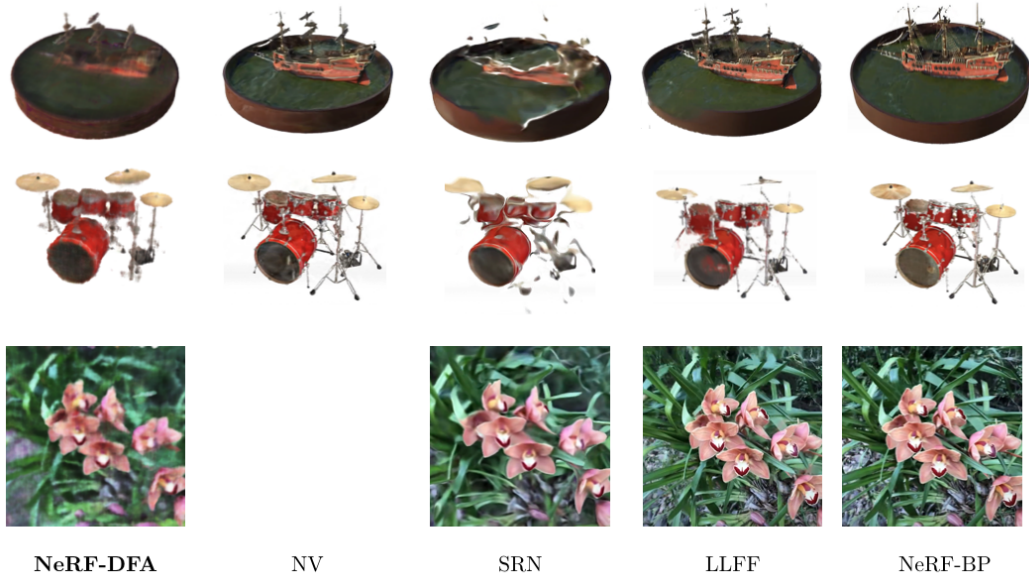

| NeRF-DFA | NV | SRN | LLFF | NeRF-BP |

Figure 1: Comparisons of NeRF-DFA with state-of-the-art methods trained with BP on the most challenging synthetic and real-world scenes. While NeRF-DFA generates render of lower quality, they maintain multi-view consistency and exhibit no geometric artefacts. BP results from [39].

**Setting** We employ Neural Radiance Fields (NeRF) [39], the state-of-the-art for neural view synthesis. NeRF represents scenes as a continuous 5D function of space–three spatial coordinates, two viewing angles–and outputs a point-wise RGB radiance and opacity. A ray-casting renderer can then query the network to generate arbitrary views of the scene. The network modeling the continuous function is 10 layers deep. Two identical networks are trained: the *coarse* network predictions inform the renderer about the spatial coordinates that the *fine* network should preferentially evaluate to avoid empty space and occluded regions.

**Results** We report quantitative results of training NeRF with DFA in Table 1. Neural view synthesis methods are often better evaluated qualitatively: we showcase some renders in Figure 1.

On a dataset of renders featuring complex scenes with non-Lambertian materials (NeRF-Synthetic [39]), NeRF-DFA outperforms two previous fine-tuned state-of-the-art methods–Scene Representation Networks (SRN) [38] and Local Light Field Fusion (LLFF) [35]–and nearly matches the performance of Neural Volumes (NV) [37]. While DFA underperforms alternative methods trained with BP on the real world view dataset (LLFF-Real [35]), its performance remains significant: real world view synthesis is a challenging tasks, and this level of PSNR indicates that learning is indeed happening.

In particular, we find that NeRF-DFA retains the key characteristics of NeRF-BP: it can render view-dependant effects, and is multi-view consistent. The last point is an especially important achievement, and most visible in the video linked in appendix E, as it is a challenge for most algorithms [33–35, 38]. The main drawback of NeRF-DFA appears to be a seemingly lower render definition. The

Table 1: Peak Signal to Noise Ratio (PSNR, higher is better) of neural view synthesis methods trained with backpropagation against NeRF trained with DFA. Even when trained with DFA, NeRF outperforms two state-of-the-art methods on a synthetic dataset (NeRF-Synthetic), and achieves fair performance on a challenging real world views datasets (LLFF-Real). BP results from [39].

|  | NV | SRN | LLFF | NeRF | |
|  | BP | BP | BP | BP | DFA |
|---|---|---|---|---|---|
| **NeRF-Synthetic** | 26.05 | 22.26 | 24.88 | 31.01 | 25.41 |
| **LLFF-Real** | / | 22.84 | 24.13 | 26.50 | 20.77 |

NeRF architecture has not been fine-tuned to achieve these results: DFA works out-of-the-box on this advanced method. Future research focusing on architectural changes to NeRF could improve performance with DFA; some preliminary results are included in appendix E of the supplementary.

### 3.1.2  Click-through rate prediction with recommender systems

We have demonstrated that DFA can train large fully connected networks on the difficult task of neural view synthesis. We now seek to use DFA in more complex heterogeneous architectures, combining the use of fully connected networks with other machine learning methods. *Recommender systems* are an ideal application for such considerations.

**Background**  Recommender systems are used to model the behavior of users and predict future interactions. In particular, in the context of click-through rate (CTR) prediction, these systems model the probability of a user clicking on a given item. Building recommender systems is hard [41]: their input is high-dimensional and sparse, and the model must learn to extract high-order combinatorial features from the data. Moreover, they need to do so efficiently, as they are used to make millions of predictions and the training data may contain billions of examples.

Factorization Machines (FM) [42] use inner-products of latent vectors between features to extract pairwise feature interactions. They constitute an excellent baseline for shallow recommender systems, but fail to efficiently transcribe higher-level features. To avoid extensive feature engineering, it has been suggested that deep learning can be used in conjunction with wide shallow models to extract these higher-level features [43]. In production, these systems are regularly retrained on massive datasets: the speedup allowed by backward unlocking in DFA is thus of particular interest.

**Setting**  Deep Factorization Machines (DeepFM) [44] combine FM and a deep fully connected neural network, which we train with DFA. The input embedding is still trained directly via gradient descent, as weight transport is not necessary to backpropagate through the FM. Deep & Cross Networks (DCN) [45] replace the FM with a Cross Network, a deep architecture without non-linearities capable of extracting high-degree interactions across features. We train the fully connected network, the deep cross network, and the embeddings with DFA. Finally, Adaptative Factorization Network (AFN) [46] uses Logarithmic Neural Networks [47] to enhance the representational power of its deep component. We evaluate these methods on the Criteo dataset [48], which features nearly 46 million samples of one million sparse features. This is a difficult task, where performance improvements of the AUC on the *0.001-level* can enhance CTR significantly [43].

**Results**  Performance metrics are reported in Table 2. To obtain these results, a simple hyperparameter grid search over optimization and regularization parameters was performed for BP and DFA independently. DFA successfully trains all methods above the FM baseline, and in fact matches BP performance in both DeepFM and AFN. Because of their complexity, recommender systems require intensive tuning and feature engineering to perform at the state-of-the-art level–and reproducing existing results can be challenging [49]. Hence, it is not surprising that a performance gap exists with Deep&Cross–further fine-tuning may be necessary for DFA to reach BP performance.

Alignment measurements corroborate that learning is indeed occurring in the special layers of Deep&Cross and AFN–see appendix A of the supplementary for details. Our results on recommender systems support that DFA can learn in a large variety of settings, and that weight transport is not necessary to solve a difficult recommendation task.

Table 2: AUC (higher is better) and log loss (lower is better) of recommender systems trained on the Criteo dataset [48]. Even in complex heterogeneous architectures, DFA performance is in line with BP. Values in **bold** indicate DFA AUC within 0.001 from the BP AUC or better.

|  | FM | DeepFM | | Deep&Cross | | AFN | |
|---|---|---|---|---|---|---|---|
|  |  | BP | DFA | BP | DFA | BP | DFA |
| AUC | 0.7915 | 0.7954 | **0.7956** | 0.8104 | 0.8009 | 0.7933 | **0.7924** |
| Loss | 0.4687 | 0.4610 | **0.4624** | 0.4414 | 0.4502 | 0.4630 | **0.4621** |

## 3.2 Geometric Learning with Graph Convolutional Networks

The use of sophisticated architectures beyond fully connected layers is necessary for certain tasks, such as *geometric learning* [50], where information lies in a complex structured domain. To address geometric learning tasks, methods capable of handling graph-based data are commonly needed. Graph convolutional neural networks (GCNNs) [51–54] have demonstrated the ability to process large-scale graph data efficiently. We study the applicability of DFA to these methods, including recent architectures based on an attention mechanism. Overall, this is an especially interesting setting, as DFA fails to train more classic 2D image convolutional layers [24].

**Background** Complex data like social networks or brain connectomes lie on irregular or non-Euclidean domains. They can be represented as graphs, and efficient processing in the spectral domain is possible. Non-spectral techniques to apply neural networks to graphs have also been developed [55–57], but they exhibit unfavorable scaling properties. The success of CNNs in deep learning can be attributed to their ability to efficiently process structured high-dimensional data by sharing local filters. Thus, a generalization of the convolution operator to the graph domain is desirable: [51] first proposed a spectral convolution operation for graphs, and [52] introduced a form of regularization to enforce spatial locality of the filters. We use DFA to train different such GCNNs implementations. We study both spectral and non-spectral convolutions, as well as methods inspired by the attention mechanism. We consider the task of semi-supervised node classification: nodes from a graph are classified using their relationship to other nodes as well as node-wise features.

**Setting** Fast Localized Convolutions (ChebConv) [53] approximate the graph convolution kernel with Chebyshev polynomials, and are one of the first scalable convolution methods on graph. Graph Convolutions (GraphConv) [54] remove the need for an explicit parametrization of the kernel by enforcing linearity of the convolution operation on the graph Laplacian spectrum. It is often considered as the canonical graph convolution. More recent methods do not operate in the spectral domain. Spline Convolutions (SplineConv) [58] use a spline-based kernel, enabling the inclusion of information about the relative positioning of nodes, enhancing their representational power–for instance in the context of 3D meshes. Graph Attention Networks (GATConv) [59] use self-attention [60] layers to enable predictions at a given node to *attend* more specifically to certain parts of its neighborhood. Finally, building upon Jumping Knowledge Network [61], Just Jump (DNAConv) [62] uses multi-head attention [63] to enhance the aggregation process in graph convolutions and enable deeper architectures. Note our implementation of DFA allows for limited weight transport within attention – see appendix D. We use PyTorch Geometric [64] for implementation of all of these methods. We evaluate performance on three citation network datasets: Cora, CiteSeer, and PubMed [65].

**Results** We report classification accuracy in Table 3. BP and DFA regularization and optimization hyperparameters are fine-tuned separately on the Cora dataset. In general, we find that less regularization and lower learning rates are needed with DFA. DFA successfully trains all graph methods, independent of whether they use the spectral domain or not, and even if they use attention. Furthermore, for GraphConv, SplineConv, and GATConv DFA performance nearly matches BP.

As GCNNs struggle with learning meaningful representations when stacking many layers [66], all architectures but DNAConv are quite shallow (two layers). However, DFA performance is still significantly higher than that of a shallow training method–see appendix B for details. The lower performance on DNAConv is not a failure to learn: alignment measurements in appendix A show that

Table 3: Classification accuracy (%, higher is better) of graph convolution methods trained with BP and DFA, on citation networks [65]. But for ChebConv and DNAConv, DFA performance nearly matches BP performance. Values in **bold** when DFA is within 2.5% of BP.

|  | ChebConv | | GraphConv | | SplineConv | | GATConv | | DNAConv | |
|---|---|---|---|---|---|---|---|---|---|---|
|  | BP | DFA | BP | DFA | BP | DFA | BP | DFA | BP | DFA |
| **Cora** | 79.2 | 75.4 | 80.1 | **79.9** | 81.0 | 77.7 | 82.6 | **80.6** | 84.6 | **82.9** |
| **CiteSeer** | 69.5 | **67.6** | 71.6 | **69.4** | 70.0 | **69.8** | 72.0 | **71.2** | 73.4 | 70.8 |
| **PubMed** | 79.5 | 75.7 | 78.8 | **77.8** | 77.5 | **77.2** | 77.7 | **77.1** | 87.2 | 79.9 |

|          |     | GAE   |       |
|----------|-----|-------|-------|
|          |     | BP    | DFA   |
| **Cora** | AUC | 0.918 | 0.900 |
|          | AP  | 0.918 | 0.900 |
| **CiteSeer** | AUC | 0.886 | 0.879 |
|          | AP  | 0.895 | 0.889 |
| **PubMed** | AUC | 0.967 | 0.945 |
|          | AP  | 0.966 | 0.945 |

Table 4: AUC and Average Precision (AP, higher is better) for a Graph-Conv GAE trained with BP or DFA on citation networks. DFA reproduces BP performance.

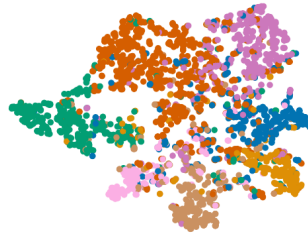

Figure 2: t-SNE visualization of the hidden layer activations of a two-layer GraphConv trained on Cora with DFA. Classes forms clear clusters, indicating that a useful intermediary representation is learned. Colors represent different classes.

learning is indeed occurring. It may be explained instead by a need for more in-depth fine-tuning, as this is a deep architecture with 5 successive attention layers.

We further demonstrate that DFA helps graph convolutions learn meaningful representations by aplying t-SNE [67, 68] to the hidden layer activations in GraphConv (Figure 2). Cluster of classes are well-separated, indicating that a useful intermediary representation is derived by the first layer.

**Graph autoencoders**    We consider one last application of graph convolutions, in the context of graph autoencoders (GAE). We train a non-probabilistic GAE [69] based on GraphConv with DFA, and report results in Table 4. DFA performance is always in line with BP.

### 3.3   Natural Language Processing with Transformers

We complete our study by training a Transformer [63] on a language modelling task. Transformers have proved successful in text, image, music generation, machine translation, and many supervised NLP tasks [63, 70–73]. Here, we demonstrate that DFA can train them, and we show the influence of tuning the optimizer hyperparameters in narrowing the gap with BP.

**Background**    NLP has largely benefited from advances in deep learning. Recurrent Neural Networks were responsible for early breakthroughs, but their sequential nature prevented efficient parallelization of data processing. Transformers are attention-based models that do not rely on recurrence or convolution. Their ability to scale massively has allowed the training of models with several billion parameters [74, 75], obtaining state-of-the-art results on all NLP tasks: Transformers now top the prominent SQuAD 2.0 [76, 77] and SuperGLUE [78] benchmarks. In parallel, transfer learning in NLP has leaped forward thanks to language modelling, the unsupervised task of predicting the next word. It can leverage virtually unlimited data from web scraping [79]. This enabled the training of *universal language models* [80] on extremely large and diversified text corpora. These models are useful across a wide range of domains, and can solve most NLP tasks after fine-tuning.

**Setting**    The prominence of both language modelling and Transformers gives us the ideal candidate for our NLP experiments: we train a Transformer to predict the next word on the WikiText-103 dataset [81], a large collection of *good* and *featured* Wikipedia articles. We use byte-pair-encoding [82] with 32,000 tokens. We adopt a Generative Pre-Training (GPT) setup [70]: we adapt the Transformer, originally an encoder-decoder model designed for machine translation, to language modelling. We keep only the encoder and mask the tokens to predict. Our architecture consists in 6 layers, 8 attention heads, a model dimension of 512, and a hidden size of 2048 in the feed-forward blocks. The text is sliced in chunks of 128 tokens and batches of 64 such chunks, resulting in 8192 tokens per batch. Our baseline is trained with BP using the optimization setup of [63]. We found perplexity after 20 epochs to be an excellent indicator of perplexity at convergence; to maximize the number of experiments we could perform, we report the best validation perplexity after 20 epochs. We study two ways of implementing DFA: applying the feedback after every encoder block (*macro*) or after every layer in

Table 5: Best validation perplexity after 20 epochs of a Transformer trained on WikiText-103 (lower is better). The BP and DFA baselines share all hyper-parameters. In *Macro* the feedback is applied after every transformer layer, implying weight transport at the layer-scale, while in *Micro* the feedback is applied after every sub-layer. The learning rate of Adam without the learning rate scheduler is $5.10^{-5}$. With the scheduler, the initial learning rate is $1.10^{-4}$ and it is multiplied by 0.2 when performance plateaus, with a patience of 1.
* *score after 22 epochs to let the learning rate scheduler take effect*

|  | **DFA** | | | | **BP** | |
|  | Baseline | + Adam | $+ \beta_2 = 0.999$ | + LR schedule | Baseline | $+ \beta_2 = 0.999$ |
| --- | --- | --- | --- | --- | --- | --- |
| **Macro** | 95.0 | 77.1 | 55.0 | 52.0 | 34.4 | 29.8 |
| **Micro** | 182 | 166 | 99.9 | 93.3* | | |

those blocks (*micro*). The *macro* setting enables weight transport at the block-scale, and some weight transport remain in the *micro* setting as well: to train the input embeddings layer, by backpropagation through the first encoder block, and for the values matrices in attention – see Appendix D for details.

**Results** Our results are summarized in Table 5. Hyper-parameters fine-tuned for BP did not fare well with DFA, but changes in the optimizer narrowed the gap between BP and DFA considerably. The learning rate schedule used on top of Adam [83] in [63] proved detrimental. Using Adam alone required reducing the learning rate between BP and DFA. Increasing $\beta_2$ from 0.98 [63] to 0.999 improved performance significantly. Finally, a simple scheduler that reduces the learning rate when the validation perplexity plateaus helped reducing it further. Considering that the perplexity of the shallow baseline is over 400, DFA is clearly able to train Transformers. However, our results are not on par with BP, especially in the *micro* setting. A substantial amount of work remains to make DFA competitive with BP, even more so in a minimal weight transport scenario. The large performance improvements brought by small changes in the optimizer indicate that intensive fine-tuning, common in publications introducing state-of-the-art results, could close the gap between BP and DFA.

## 4   Conclusion and outlooks

We conducted an extensive study demonstrating the ability of DFA to train modern architectures. We considered a broad selection of domains and tasks, with complex models featuring graph convolutions and attention. Our results on large networks like NeRF and Transformers are encouraging, suggesting that with further tuning, such leading architectures can be effectively trained with DFA. Future work on principled training with DFA–in particular regarding the influence of common practices and whether new procedures are required–will help close the gap with BP.

More broadly, we verified for the first time that learning under synaptic asymmetry is possible beyond fully-connected layers, and in tasks significantly more difficult than previously considered. This addresses a notable concern in biologically-plausible architectures. DFA still requires an implausible global feedback pathway; however, local training has already been demonstrated at scale. The next step towards biologically-compatible learning is a local method without weight transport.

While the tasks and architectures we have considered are not biologically inspired, they constitute a good benchmark for *behavioural realism* [20]. Any learning algorithm claiming to approximate the brain should reproduce its ability to solve complex and unseen task. Furthermore, even though the current implementation of mechanisms like attention is devoid of biological considerations, they represent broader concepts applicable to human brains [84]. Understanding how our brain learns is a gradual process, and future research could incorporate further realistic elements, like spiking neurons.

Finally, unlocking the backward pass in large architectures like Transformers is promising. More optimized implementation of DFA–built at a lower-level of existing ML libraries–could unlock significant speed-up. Leveraging the use of a single random projection as the cornerstone of training, dedicated accelerators may employ more exotic hardware architectures. This will open new possibilities in the asynchronous training of massive models.

## Broader Impact

**Of our survey** This study is the first experimental validation of DFA as an effective training method in a wide range of challenging tasks and neural networks architectures. This significantly broadens the applications of DFA, and more generally brings new insight on training techniques alternative to back-propagation. From neural rendering and recommender systems, to natural language processing or geometric learning, each of these applications has its own potential impact. Our task selection process was motivated by current trends in deep learning, as well as by technically appealing mechanisms (graph convolutions, attention). A limit of our survey is that our–arguably biased–selection of tasks cannot be exhaustive. Our experiments required substantial cloud compute resources, with state-of-the-art GPU hardware. Nevertheless, as this study provides new perspectives for hardware accelerator technologies, it may favor the application of neural networks in fields previously inaccessible because of computational limits. Future research on DFA should continue to demonstrate its use in novel contexts of interest as they are discovered.

**Of the considered applications** Each of the applications considered in our study has a wide potential impact, consider for example the impact of textual bias in pretrained word embeddings [85]. We refer to [86] and references therein for a discussion of ethical concerns of AI applications.

**Of DFA as a training method** DFA enables parallelization of the backward pass and places a single operation at the center of the training process, opening the prospect of reducing the power consumption of training chips by an order of magnitude [31]. Not only is more efficient training a path to more environmentally responsible machine learning [87], but it may lower the barrier of entry, supporting equality and sustainable development goals. A significant downside of moving from BP to DFA is a far more limited understanding of how to train models and how the trained models behave. There is a clear empirical understanding of the impact of techniques such as batch normalization or skip connections on the performance of BP; new insights need to be obtained for DFA. BP also enjoys decades of works on topics like adversarial attacks, interpretability, and fairness. Much of this work has to be cross-checked for alternative training methods, something we encourage further research to consider as the next step towards safely and responsively scaling up DFA.

**Of biologically motivated methods** Finally, a key motivation for this study was to demonstrate that learning challenging tasks was possible without weight transport. Biologically motivated methods are a more foundational research direction, and as such the possible long-term impact of our findings is harder to estimate under this light. However, fundamental research of this kind is important to open new pathways for ML and neuroscience.

## Acknowledgments and Disclosure of Funding

We thank Igor Carron and Laurent Daudet for the general guidance on the subject of this investigation and the insightful comments, as well as the larger LightOn team for their support. We also thank the anonymous reviewers for their useful comments.

Florent Krzakala acknowledges support by the French Agence Nationale de la Recherche under grants ANR17-CE23-0023-01 PAIL and ANR-19-P3IA-0001 PRAIRIE; additional funding is acknowledged from "Chaire de recherche sur les modèles et sciences des données", Fondation CFM pour la Recherche.

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
