[Supplementary Material]

# Appendix

We first provide additional elements to corroborate our findings: alignment measurement (Section A), and shallow baselines (Section B). We then discuss the process of adapting the considered architectures for DFA (Section C), and the issue of weight transport in attention layers (Section D). We provide some supplementary results for NeRF (Section E), including details of performance on each scene of each datatset, and a discussion on possible mitigation of DFA shortcomings. Finally, we outline steps necessary for reproduction of this work (Section F).

## A   Alignment

**Alignment measurement**   In feedback alignment methods, the forward weights learn to *align* with the random backward weights, making the delivered updates useful. This alignment can be quantified by measuring the cosine similarity between the gradient signal delivered by DFA $\mathbf{B}_i \delta \mathbf{a}_y$ and the gradient signal BP would have delivered $\mathbf{W}_{i+1}^T \delta \mathbf{a}_{i+1}$. For learning to occur and DFA to work as a training method, there must be alignment. This can be measured numerically [24]. Measuring alignments allows to check whether or not the layers are effectively being trained by DFA, regardless of performance metrics. We note that any alignment value superior to 0 signifies that learning is occuring. Values closer to 1 indicate a better match with BP, but small alignment values are sufficient to enable learning. We report values measured at the deepest DFA layer.

**Recommender systems**   We measure alignment on the Criteo dataset, in the two architectures featuring non-conventional fully-connected layers: Deep & Cross and AFN. Alignment is measured after 15 epochs of training, and averaged over a random batch of 512 samples. Results are reported in table A.1. These alignment measurements indicate that learning is indeed occurring in the cross and logarithmic layers. High-variance of alignment in the cross layers is unique: it may be explained by the absence of non-linearity, and account for the difference in performance between BP and DFA on this architecture–which is higher than on the others.

Table A.1: Alignment cosine similarity (higher is better, standard deviation in parenthesis) of recommender systems as measured on the Criteo dataset. Learning occurs in both architectures, and high variance may explain the larger performance gap on Deep & Cross compared to other methods.

|  | Deep & Cross | AFN |
|---|---|---|
| **Alignment** | 0.40 (0.91) | 0.49 (0.08) |

**Graph convolutions**   We measure alignment on the Cora dataset, after 250 epochs of training, averaging values over every sample available–train, validation, and test split included. Results are reported in Table A.2. We observe high alignment values in all architectures, indicative that learning is indeed occuring. Slightly lower values in SplineConv and GATConv may be explained by the use of the Exponential Linear Unit (ELU) instead of the Rectified Linear Unit (ReLU) used as activation in other architectures.

Table A.2: Alignment cosine similarity (standard deviation in parenthesis) of various graph convolutions architectures as measured on the Cora dataset. These values corroborate that DFA successfully trains all architectures considered.

|  | ChebConv | GraphConv | SplineConv | GATConv | DNAConv |
|---|---|---|---|---|---|
| **Alignment** | 0.87 (0.12) | 0.77 (0.25) | 0.56 (0.22) | 0.63 (0.18) | 0.92 (0.30) |

## B   Shallow baselines

**Shallow learning**   We compare DFA to BP, but also to shallow learning–where only the topmost layer is trained. While DFA may not reach the performance level of BP, it should still vastly

| BP | DFA | SHALLOW |

Figure A.1: Comparisons of Tiny-NeRF trained with BP, DFA, and a shallow approach. Shallow training is insufficient to learn scene geometry. Lego scene from the NeRF synthetic dataset.

outperform shallow learning: failure to do so would mean that the weight updates delivered by DFA are useless. On a simple task like MNIST, a shallow baseline may be as high as 90%. However, given the difficulty of the tasks we consider, the shallow baseline is here usually much lower.

**NeRF**    Because NeRF models are expensive to train–up to 15 hours on a V100–we consider a simplified setup for the shallow baseline, NeRF-Tiny. This setup operates at half the full resolution of the training images available, runs for 5000 iterations only, and does away with view-dependant characteristics. Furthermore, the network is cut down to 3 layers of half the width of NeRF, and no coarse network is used to inform the sampling. We train this network on the Lego scene of the NeRF-Synthetic dataset, and compare results.

Figure A.1 presents renders generated by NeRF-Tiny trained with BP, DFA, and a shallow approach. While BP and DFA delivers similar renders, shallow training fails to reproduce even basic scene geometry, instead outputting a diffuse cloud of colors. This highlights that while DFA may not reach a level of performance on-par with BP on NeRF, it nonetheless delivers meaningful updates enabling the learning of complex features.

**Recommender systems**    Because recommender systems require fine-tuning, we perform the same hyperparameter search for shallow learning than for DFA and BP. Results are detailed in Table A.3. Performance of shallow training is always well under BP and DFA–remember that *0.001-level* matter in recommender systems. In particular, in Deep & Cross, where there was the biggest gap between BP and DFA, the performance of the shallow method is extremely poor, well below the FM baseline. Finally, it is expected to see that DeepFM recovers more or less the performance of FM even with a shallow baseline.

Table A.3: Shallow baseline for recommender system models on the Criteo dataset. Performance is always well below BP and DFA, as expected.

|  | DeepFM | Deep&Cross | AFN |
|---|---|---|---|
| AUC | 0.7920 | 0.7324 | 0.7859 |
| Loss | 0.4682 | 0.5010 | 0.4685 |

**Graph convolutions**    We use the same hyperparameters as for DFA to produce the shallow baseline on graph datasets. Results are reported in Table A.4. Performance is always much worse than BP and DFA. GATConv recovers the best performance: random attention layers may still deliver useful features [88], as do random convolutions.

We also produce t-SNE visualizations of the hidden layer activations for a BP-trained network and a shallow-trained one (Figure A.2). t-SNE hyperparameters are identical between all three visualizations. In the shallow case, the hidden layer is not trained at all and remained in its random initialized state: in this case, t-SNE is unable to extract any structure.

Figure A.2: t-SNE visualization of the hidden layer activations of a two-layer GraphConv trained on Cora with a shallow approach, BP, and DFA. The shallow approach does not train the hidden layer and t-SNE fail to extract any information from the randomly initialized layer. DFA and BP visualizations show identical level of separation between clusters.

Table A.4: Shallow baseline for GCNNs on Cora, CiteSeer, and PubMed [65]. Performance is always well below BP and DFA.

|  | ChebConv | GraphConv | SplineConv | GATConv | DNAConv |
|---|---|---|---|---|---|
| **Cora** | 23.3 | 37.0 | 39.6 | 59.4 | 30.2 |
| **CiteSeer** | 27.4 | 33.8 | 30.1 | 49.8 | 24.0 |
| **PubMed** | 37.6 | 44.8 | 44.2 | 67.8 | 42.2 |

**Transformers** In the baseline setting (optimizer and hyper-parameters of [63]), a Transformer trained in the shallow regime yields a perplexity of 428 on WikiText-103. We do not consider other settings, as the cost of training a Transformer is high and we do not expect any meaningful improvements–as with NeRF above.

## C  Adapting architectures to DFA

In general, our implementation of DFA for all architecture follows the spirit of the original paper [19]. We introduce a random feedback $\mathbf{B}\delta\mathbf{a}_y$ after every non-linearity, and do not use any architecture-specific structure or operation to build the feedback. For graphs and transformers, we do share the backward random matrix for all nodes in a graph and for all tokens in a sentence. This is not only more computationally efficient, but also necessary for proper training: if the random matrix was different for each node/token, the graph/attention layers would receive incoherent feedbacks coming from different random matrices and alignment would be impossible. Finally, our global feedback matrix is initialized from $\mathcal{U}(-1, 1)$ and normalized with the square root of the output dimension of every layer.

**NeRF** We use an architecture identical to the one used in [39], but based on the effective code implementation rather than the description in the paper[1]. During our tests, we have found that lowering the learning rate to $1.10^{-4}$ rather than $5.10^{-4}$ works best with DFA.

**Recommender systems** For all training methods (BP, DFA, and shallow), we have conducted independent hyperparameter searches. We performed a grid search over the learning rate, from $1.10^{-4}$ to $1.10^{-3}$ in $1.10^{-4}$ steps, as well as over the dropout probability, from 0.1 to 0.5 in 0.1 steps (where applicable). On DeepFM, this search leads to reduce the learning rate from $3.10^{-4}$ with BP to $5.10^{-5}$ with DFA, but to keep the 0.5 dropout rate. On Deep & Cross, we reduce learning rate from $2.10^{-4}$ to $5.10^{-5}$, with no dropout in both cases. In AFN, we reduce dropout from $4.10^{-4}$ to $3.10^{-4}$ and dropout from 0.3 to 0.

**Graph convolutions** We manually test for a few hyperparameters configuration on the Cora dataset, focusing on learning rate, weight decay, and dropout. We do not consider architectural changes, such

as changing the number of filters or of attention heads. For ChebConv and GraphConv, we reduce weight decay to $1.10^{-4}$ instead of $5.10^{-4}$, and set the dropout rate to 0 and 0.1 respectively, instead of 0.5 with BP. For SplineConv, we find that no change in the hyperparameters are necessary. For GATConv, we reduce weight decay to $1.10^{-4}$ instead of $5.10^{-4}$ and reduce dedicated dropout layer to 0.1 instead of 0.6 but keep the 0.6 dropout rate within the GAT layer. Finally, on DNAConv we disable weight decay entirely, instead of an original value of $5.10^{-4}$, double the learning rate from $5.10^{-3}$ to $1.10^{-2}$, and disable dropout entirely. In all cases, we share the backward random matrix across all nodes in a graph.

**Transformers** The model hyper-parameters were fixed across all of our experiments, except for the number of attention heads in one case, that we will precise below, and dropout. We tested several values of dropout probability between 0 and 0.5, but found the original value of 0.1 to perform best. We manually tested a number of optimizers, optimizer parameters and attention mechanisms. We tested four combinations of optimizers and schedulers : Adam with the scheduler used in [63], Adam alone, RAdam [89] alone, and Adam with a scheduler that reduces the learning rate when the validation perplexity plateaus. We found it necessary to reduce the initial learning rate of Adam from $1.10^{-4}$ to $5.10^{-5}$, although it could be set back to $1.10^{-4}$ with a scheduler. We tried two values of $\beta_2$: 0.98 and 0.999. We also tried to change $\beta_1$ and observed some small differences that were not significant enough for the main text. Finally, we tried three attention mechanisms in addition to the standard multihead scaled dot-product attention: the dense and random (learnable) Synthesizers of [88], as well as the fixed attention patterns of [90]. The latter needed to be adapted to language modelling to prevent attending to future tokens, which led us to reduced the number of attention heads to 4. The backward random matrix is always shared across all tokens and batches.

# D   Weight transport and attention

We consider an attention layer operating on input $\mathbf{x}$. The queries, keys, and values are respectively $\mathbf{q} = \mathbf{x}\mathbf{W}_Q; \mathbf{k} = \mathbf{x}\mathbf{W}_K; \mathbf{v} = \mathbf{x}\mathbf{W}_V$, and $d_k$ is the dimension of the queries and keys. The layer performs:

$$\text{Attention}(\mathbf{q}, \mathbf{k}, \mathbf{v}) = \text{softmax}\left(\frac{\mathbf{q}\mathbf{k}^T}{\sqrt{d_k}}\right)\mathbf{v} \tag{4}$$

When using DFA on attention, we deliver the random feedback to the top of the layer. Accordingly, to obtain updates to $\mathbf{W}_Q, \mathbf{W}_K$, and $\mathbf{W}_V$ we still to have to backpropagate through the attention mechanism itself. This involves weight transport on $\mathbf{W}_V$, sacrificing some biological realism for simplicity. Overall weight transport between layers still does not occur, and updating the layers in parallel remains possible.

Beside using FA or DFA within the attention layer, alternative mechanisms like the synthesizer [88]–which uses random attention in place of the query and key system–or fixed attention [90] can remove the need for weight transport. Implementing these mechanisms in DFA-trained Transformers, or other attention-powered architectures, will require further research.

# E   Supplementary NeRF results

**Quantitative results** We report per-scene scores for each dataset in Table A.5. BP values are taken from [39]. On three scenes of the synthetic datasets, NeRF-DFA even outperforms past state-of-the-art methods trained with BP. Note that Neural Volumes (NV) is not applicable to forward-facing view synthesis–as is required in LLFF-Real–and thus no results are reported.

**Qualitative results** We report sample renders from the NeRF-Synthetic dataset (Figure A.3) and the LLFF-Real dataset (Figure A.3), for every scene available. However, we recommend readers to consult the supplementary video[2] to make better sense of characteristics like multi-view consistency and view-dependent effects (most visible on the LLFF-Real Room scene).

**Possible future directions**   Despite retranscribing scene geometry in a multi-view consistent way, NeRF produces renders of a lower quality when trained with DFA instead of BP. In particular, it struggles to transcribe small-scale details, resulting in "blurry" renders. Moreover, it displays high-frequency artefacts: not in the scene geometry, but in individual pixels taking values very distant from their neighborhood. Interestingly, this noise phenomenon is unique to NeRF-DFA: it is not observed on NeRF-BP with similar PSNR values (achieved during training) or on other methods with similar or lower PSNR. This leads us to hypothesize this is an aspect unique to DFA, possibly due to the alignment process. Indeed, DFA creates a bias on the weights, by encouraging them to be "aligned" with an arbitrary values dependant on the random matrix used. It is possible this could introduce random noise in the final renders–though we leave a more principled experiment to future research.

To attempt to alleviate this issue, we first consider NeRF-Dual. In NeRF-Dual, we average the pixel-wise prediction between the fine and coarse network, to attempt to remove some of the noise. To do so, we first still use the coarse network to create a probability distribution for the hierarchical sampling. Then, we evaluate again both the coarse and fine networks at the locations informed by this probability distribution. Compared to vanilla NeRF, this requires an extra batch of evaluation of the coarse network for all rays–rougly speaking, this increases inference time by 30-50% depending on the coarse network architecture considered. We note that this is not applied during training, so that training times remain identical.

Figure A.3 and Figure A.4 showcase comparisons between NeRF and NeRF-Dual trained with DFA on all scenes. When viewed at high resolution–such as in our supplementary video–the NeRF-Dual renders are more pleasing, especially for the full scenes. They remove most of the high-frequency noise, leading to smoother renders. However, this averaging process further blurs small-scale details in the render. This is especially visible in the NeRF-Synthetic dataset, on scenes like Ficus. Furthermore, NeRF-Dual introduces novel artefacts in the Mic and Ship scenes, with areas improperly colored with a violet tint. The cause for these artefacts is unknown, but they show that NeRF-Dual is far from a silver bullet. The PSNR is also minimally increased, by less than 0.5 per scene. Nevertheless, this shows some promise in possibilities to allieviate the shortcomings of NeRF-DFA. It is possible that changes to the overall rendering process, or the use of classic image processing techniques, may help enhance the NeRF-DFA images.

Table A.5: Per-scene PSNR for NeRF DFA and BP against other state-of-the-art methods on the Nerf-Synthetic and LLFF-Real. DFA performance is fairly homogeneous across each dataset and in line with the differences in other methods.

|  | NV | SRN | LLFF | NeRF | |
|---|---|---|---|---|---|
|  | BP | BP | BP | BP | DFA |
| **NeRF-Synthetic** | **26.05** | **22.26** | **24.88** | **31.01** | **25.41** |
| Chair | 28.33 | 26.96 | 28.72 | 33.00 | 28.74 |
| Drums | 22.58 | 17.18 | 21.13 | 25.01 | 22.15 |
| Ficus | 24.79 | 20.73 | 21.79 | 30.13 | 25.61 |
| Hotdog | 30.71 | 26.81 | 31.41 | 36.18 | 28.03 |
| Lego | 26.08 | 20.85 | 24.54 | 32.54 | 24.93 |
| Materials | 24.22 | 18.09 | 20.72 | 29.62 | 25.15 |
| Mic | 27.78 | 26.85 | 27.48 | 32.91 | 25.43 |
| Ship | 23.93 | 20.60 | 23.22 | 28.65 | 23.25 |
| **LLFF-Real** | | **22.84** | **24.13** | **26.50** | **20.77** |
| Room | | 27.29 | 28.42 | 32.70 | 24.20 |
| Fern | | 21.37 | 22.95 | 25.17 | 21.82 |
| Leaves | | 18.24 | 19.52 | 20.92 | 16.50 |
| Fortress | | 26.63 | 29.40 | 31.16 | 25.16 |
| Orchids | | 17.37 | 18.52 | 20.36 | 16.73 |
| Flower | | 26.63 | 25.46 | 27.40 | 21.55 |
| T-Rex | | 22.87 | 24.15 | 26.80 | 19.43 |
| Horns | | 24.33 | 24.70 | 27.45 | 20.75 |

Finally, we also experimented with increasing the capacity of the fine network, by widening its layers to 512 neurons. We call this architecture NeRF-XL. However, we have not succeeded in getting PSNR values higher than with vanilla NeRF on DFA. In particular, the training process becomes much more cumbersome, as multi-GPU parallelism is needed to fit the model. It is possible that higher network capacity may help learning both the task at hand and to align simultaneously, but further work is required.

## F  Reproducibility

**Hardware used**    All main experiments require at most a single NVIDIA V100 GPU with 16GB of memory to reproduce. Alignment measurement on large architectures (NeRF and Transformers) require a second identical GPU to keep a copy of the network to evaluate BP gradients.

We estimate that a total of around 10,000 GPU-hours on V100s were necessary for this paper. Accordingly, we estimate the cloud-computing carbon impact of this paper to be of 1700 $kgCO_2eq$[3].

However, without hyperparameter searches, our results can be reproduced with less than 500 GPU-hours on V100s, with most of that budget going to NeRF and Transformers.

**Implementation**    We use the shared random matrix trick from [24] to reduce memory use in DFA and enable its scaling to large networks. We use PyTorch [91] for all experiments. For reference implementation of the methods considered, we relied on various sources. Our NeRF implementation is based on the PyTorch implementation by Krishna Murthy[4], with modifications to allow for proper test and validation, as well as DFA and multi-GPU support. For recommender systems, we use the `torchfm` package[5]. Finally, we use PyTorch Geometric [64] for all graph operations. Our Transformer implementation is our own. Our code is available as supplementary material.

**NeRF**    We provide training, testing, and rendering code along with the configurations used to obtain our results. An example to reproduce our results is given in the supplementary code repository. Given the computing cost associated with training a NeRF, we also provide our trained models.

**Recommender systems**    We provide bash scripts to reproduce the results in Table 2 and A.3, with the results of our hyperparameter search. We provide code to reproduce the results in Table A.1.

**Graph convolutions**    We provide the code to reproduce all of our results. Note that the t-SNE results are not exactly reproducible, as the CUDA implementation used is non-deterministic.

**Transformers**    We provide bash scripts to reproduce Table 5 and the shallow results.

**NeRF**  **NeRF-Dual**  **NeRF**  **NeRF-Dual**

Figure A.3: Sample renders for every scene of the NeRF-Synthetic dataset, for NeRF and NeRF-Dual trained with DFA.

| NeRF | NeRF-Dual | NeRF | NeRF-Dual |
|------|-----------|------|-----------|

Figure A.4: Sample renders for every scene of the LLFF-Real dataset, for NeRF and NeRF-Dual trained with DFA.

## Footnotes

[1] `https://github.com/bmild/nerf/issues/11`

[2]`https://www.youtube.com/watch?v=sinch7013LY`

[3]`https://mlco2.github.io/impact#compute`

[4]`https://github.com/krrish94/nerf-pytorch`

[5]`https://github.com/rixwew/pytorch-fm`