[Reviews · NeurIPS 2020]

Review 1

Summary and Contributions: This paper revisits the direct feedback alignment algorithm, benchmarking it on a wider variety of datasets than had been before. The author finds that, despite previous results suggesting that DFA scales poorly to difficult image classification problems, it works well on a variety of other tasks that had not been considered before.

Strengths: This is a paper focused on empirical results, aand the experiments are extensive and thorough, with appropriate attention given to the need for using different hyperparameters for backprop and DFA.

Weaknesses: 1. The choice of tasks in the paper is (to this reviewer) feels arbitrary, particularly neural view synthesis and click-through rate prediction. Why did the authors choose these tasks and not others? Were the authors motivated by applications, or did they choose tasks where they thought DFA had a good chance of performing well? If so, what made these tasks seem promising? Can the authors suggests some tasks, besides image classification with ConvNets, where they would not expect DFA to perform well? 2. Related to the above, I think readers of this paper will likely wonder the following: why does DFA work on some tasks and not others? Is task or architecture the more important factor? What are the minimal changes to the problem or architecture that could "break" DFA's performance on the tasks where it works well? Could these provide insights into how to rescue DFA's performance on tasks where it has fared poorly, like image classification? 3. On the NLP task, which seem like the most "standard" task the authors tried, DFA performance lags substantially behind backprop performance. I feel this is not sufficiently emphasized in the paper. The abstract, for example, gives readers the impression that DFA works at near-backprop levels on this task, just like it does with the others.

Correctness: The empirical evaluations seem rigorously done. The claims are mostly supported by the experiments, but I feel like the current presentation is misleading in two ways. 1. As mentioned above, the NLP results are not as good as suggested by the abstract and introduction. 2. I think the title of the paper is a bit misleading. Are these tasks really a representative sample of "Modern Deep Learning Tasks and Architectures?" When I had only seen the title of this paper, I, for instance, assumed that it was going to show a way to rescue DFA performance on ImageNet. Given that it does not, and that the NLP results are not as strong as the rest, I think this paper is less about refuting the claim that DFA doesn't scale and more about showing that the situation is complicated and more study is needed. I think a different paper title could convey this impression more accurately.

Clarity: Yes, it is clearly written.

Relation to Prior Work: The work is well-motivated in the context of both previous biologically plausible learning papers and more practically minded algorithms intended to help deep learning scale better. I think more discussion of the practical implications of DFA relative to the other methods discussed would be beneficial.

Reproducibility: Yes

Additional Feedback: I think the empirical results are important and show that it is worth making a good-faith effort to scale feedback alignment-related algorithms. I would likely be willing to update my score if some of the issues I've mentioned are addressed. *************** Update: the authors have addressed several of my concerns in the response and I am raising my score accordingly. I do think acknowledging the less impressive results on the NLP task in the abstract is important (in fact, it is an interesting finding!)


Review 2

Summary and Contributions: The authors present a study that uses direct feedback alignment (DFA) to train various models on various challenging tasks. The work is motivated by arguing that DFA was so far only used on small datasets, and was shown to not perform well on computer vision tasks, in part because of the usage of CNNs in these settings. This survey challenges these views by conducting an extensive set of experiments using DFA to train s.o.t.a. models on s.o.t.a. benchmarks. The benchmarks include view synthesis, language modeling, recommender systems, and graph embedding. They compare the performance of these models to ones trained using a normal BP approach. The authors show that DFA can be competitive to classical BP in many scenarios, and also show how further improvements could be implemented. They also highlight potential benefits (e.g. parallelization) of training models with DFA vs BP. ****** Update: The author's response covers my comments and I will keep my positive score.

Strengths: The main strengths are: - Extensive set of experiments that reassess DFA - Diverse choice of s.o.t.a. benchmarks for DFA evaluation - Selecting state-of-the-art settings and benchmarks instead of the often used simple (toy) datasets. Each benchmark section has a concise set of background information, description of the setting and presentation and interpretation of the results. This makes it very clear to the reader, where DFA is competitive, and where further investigation is needed (e.g. in the NLP task).

Weaknesses: The main weaknesses are: Though the results shine a new light on DFA, the method and how it is used is not a novelty by itself, hence the study does not present a novel method or a new variation of an existing method. it 'only' applies (though very extensively) the DFA training method to existing models and existing datasets.

Correctness: The claims and the stated contributions to the field are correct. The empirical methodology and procedures (incl. grid search of hyperparameters etc.) are correct as well.

Clarity: The clarity of the paper is good. It is well structured and the different benchmarks and settings are properly explained. Most questions that came to my mind while reading were either covered directly or by the appendix material. Only critique here: some abbreviations are not introduced.

Relation to Prior Work: The authors explain how DFA was used before, what the scope of the previous experiments and their limitations were. The authors make good use of the extended bibliography and cite all the relevant previous contributions.

Reproducibility: Yes

Additional Feedback: Make sure you introduce all abbreviations.


Review 3

Summary and Contributions: This paper provides an extensive empirical evaluation of direct feedback alignment (DFA), a simple and scalable credit assignment algorithm. Unlike backpropagation of error (BP), DFA does not require weight transport and it does not need symmetric backward connectivity. Experiments are conducted on deep neural network models for neural view synthesis, recommender systems, geometric learning, and natural language processing. The experimental evidence is convincing. On the tasks considered DFA does not always match BP, but it produces useful weight updates. ** Update: I maintain my positive score after reading the authors' response.

Strengths: - Focusing on DFA was an excellent choice. Unlike feedback alignment, DFA does not require symmetric connectivity. This makes DFA a very appealing model for neuroscientists, and a useful algorithm for hardware designers (perhaps even for distributed software implementations). The feedback architecture of DFA is also a natural first choice for synthetic gradient modules (see, e.g., Lansdell et al., ICLR2020). - Large-scale experiments that involve a number of different architectural elements. - Appropriate controls.

Weaknesses: - Being a purely empirical paper, which studies an existing method, there is no theoretical or algorithmic novelty.

Correctness: - It would be good to always very clearly note in the main text whenever weight transport is violated when training with DFA (e.g., attention layers as discussed in the SM). - Eq. 1 (typo?): shouldn't $i$ start at the first layer, up to layer $N$?

Clarity: The paper is well written and clear, with sufficient detail, well positioned within the literature. I imagine it is some work, but it would be useful for the reader unfamiliar with all the architectures involved -- like me -- to provide (simple) network diagrams in the SM.

Relation to Prior Work: The paper is well-positioned in the literature. Some ideas to further strengthen the discussion section: - It could be good to cite (1), which provides more evidence that DFA has troubles in training convolutional layers in image classification tasks, and contains some surprising findings on DFA with sparse feedback. - The results could be seen as a promising starting point, which encourage improving the simple DFA baseline with a learning to learn algorithm, see e.g. (2). Personally, I find this point of view exciting.

Reproducibility: Yes

Additional Feedback:


Review 4

Summary and Contributions: The paper applies an existing algorithm called Direct Feedback Alignment (DFA) to diverse tasks and datasets, largely beyond what prior work has done experimentally with DFA. DFA is an alternative to the conventional backpropagation algorithm. The authors point out two benefits of DFA over backpropagation: 1/ DFA allows to compute the gradients of all the weights in parallel and update them synchronously, rather than successively (computational considerations). 2/ DFA does not suffer from the biologically implausible weight transport problem of backpropagation (biological considerations). The paper establishes a surprising result, that learning under synaptic asymmetry is possible beyond fully-connected layers, across a variety of tasks.

Strengths: The authors conduct an extensive study demonstrating the ability of DFA to train modern DL architectures, across a variety of tasks: neural view synthesis with neural radiance fields, click-through rate prediction recommender systems, geometric learning with graph convolutional networks, and natural language processing with transformers. This work should be useful to establish baselines for other biologically plausible learning algorithms in the future.

Weaknesses: One of the claims of the paper is that DFA can help reduce training time as well as power consumption if implemented correctly. This claim is made in the abstract, introduction, conclusion, as well as in the broader impact section. Since this claim is made in many places of the paper and used as a central argument for studying DFA, it would be helpful to have a more detailed explanation, with quantitative arguments if possible, of what would be the implications of using DFA rather than backpropagation, and what would the challenges to be overcome. With an appropriate implementation on GPUs, what are the expected gains? Denote N the number of processing stages (say N layers if we consider a standard multi layer neural net). 1/ The time required with backpropagation is: - N in the forward pass, - N in the backward pass, - plus the time required for all weight updates. 2/ One can argue that, if implemented correctly, the time required with DFA is - N in the forward pass, - 1 in the “backward pass” (all “gradients” are send through direct feedback connections in parallel), - plus the time required for all weight updates. So the overall time cut-off seems to be bounded by a factor 2. Or am I missing something? With a neuromorphic implementation, I can imagine that one would get significantly more speedup, but there seems to be many other problems to be overcome, like computing the derivatives of the forward activations (denoted f’), or the fact that you are still using backprop in the attention mechanism (as transparently explained in appendix D). These concerns should be addressed, too.

Correctness: DFA is presented in section 2. The presentation is limited to the setting of a vanilla multi-layer neural net (no skip-layer connection, no convolution, no attention mechanism, …, nothing fancy). However, in experiments, DFA is used in architectures much more complicated than this simple setting, with convolutional nets and transformers in particular. More details seem necessary to explain how this is done in these settings. I can see that appendix D briefly explains how DFA is adapted to attention mechanisms ; unfortunately, the reader has to search in the appendices by themself to find this piece of information. How does DFA adapt to other settings? Can this method be applied with any computational graph or are there limitations?

Clarity: The paper is relatively well written. It is a bit repetitive in places though.

Relation to Prior Work: The relation to prior works, which seek either for biologically plausibility or efficient use of compute resources, is well explained. Whereas previous work has applied DFA to (mostly) computer vision tasks, achieving disappointing results on challenging datasets, the authors consider here very different tasks, datasets and architectures.

Reproducibility: Yes

Additional Feedback: How are the (random) feedback weight matrices B_i (which serve as random projections) initialized? Line 263. It would be good to define beta_2. In Eq 3 you use the derivative of the activation f’ to compute the “gradients”. Is f’ necessary? If yes, it would be good to explain why. In particular, have you experimented with different activation functions or only ReLU? With ReLU, f’ is either 0 or 1, meaning that for each weight, if the postsynaptic neuron is zero then it receives zero gradient, otherwise it receives a nonzero gradient. I would expect that this is actually a critical aspect of why the method works at all. I would consider raising my score if the authors address my === After rebuttal === Thank you for answering my questions! Congrats on the nice work! I understand that with DFA we could potentially very efficiently parallelize the backward pass and thus have a speedup (Note that this speedup is unlikely more than a factor 2, since we still need to do the forward pass sequentially). It is not really clear that power consumption can be reduced though.

[Author Response · NeurIPS 2020]

We thank the reviewers for their helpful comments and feedback. We answer their questions below.

Reviewer #2 and #3 mention a lack of theoretical or algorithmic novelty as the only weakness. We believe that rigorous large-scale empirical studies are as important as the introduction of new methods and theories. We would like to highlight the novelty of our work along other axes:

- We show, for the first time, that direct feedback alignment (DFA) can match backpropagation (BP) performance on challenging tasks. This is in contrast with past studies failing to scale DFA past simple datasets like MNIST.

- Our study is unprecedented in the variety of architectures and tasks considered, the hyper-parameter tuning effort undertaken for an alternative training method, and the breadth of the controls employed. Traditionally, alternative methods are seldom evaluated on tasks beyond image recognition with convolutional networks.

- Our implementation in the supplementary material scales to large architectures and demanding tasks, solving a practical issue encountered in previous work (Bartunov et al., 2018).

**Reviewer #1**    The reviewer first questions the rationale behind our choice of tasks. Our selection was built to represent a diverse set of topics representative of modern deep learning. Neural view synthesis is a challenging subset of implicit neural representations (see the recent Sitzmann et al., 2020), an active area of research rich with potential theoretical links for future studies. Recommender systems based on click-through rate (CTR) prediction are employed at-scale in the industry, and represent a real-world deep learning scenario. Beyond considerations on the applications, we also sampled varied architectures: deep fully-connected networks (MLP) with NeRF, hybrid models combining MLP with other techniques for CTR prediction, structured networks with graph convolutions, and finally attention-based architectures. We deliberately avoided tasks involving convolutions–as DFA has already been shown to be incompatible with them. However, we did not have expectations on the performance of DFA on our selection of tasks and architectures.

The reviewer then asks why DFA performs well in some tasks and not in others. The main factor influencing the performance of DFA is the architecture. The exact mechanics explaining the successes and shortcomings of DFA is a broader topic, requiring a paper of its own. We refrain from speculation, and we hope our survey can provide inspiration for further theoretical and empirical studies of DFA, by showing that convolutions are the exception, and not the norm.

The performance of DFA lags behind backpropagation more significantly in the natural language processing (NLP) task. Indeed, the Transformer is the most complex architecture considered in our paper: it requires much more careful tuning to train well. The training of Transformers is an open topic even for BP: practices like learning rate warm-up, cosine schedule, and RAdam are fairly new (Vaswani et al., 2017, Liu et al., 2019, Popel and Bojar, 2018). Fully adapting these principles to DFA requires a substantial amount of work, beyond the scope of a single submission. We are willing to rephrase the relevant part in the abstract in the camera-ready version. However, we can not find any controversial sentence in the introduction and, as noted by Reviewer #2, we state unambiguously that there remains a clear performance gap between BP and DFA in the NLP task in lines 266-268.

**Reviewer #2**    The missing abbreviations will be added in the camera-ready version of the paper.

**Reviewer #3**    Weight transport violations only occur in the Transformer architecture. Line 259 will be clarified and a mention of weight transport violation will be added in the legend of Table 5 in the camera-ready version. Also, we will include citations to Landsell et al. and Crafton et al. and fix the typo in equation 1. Finally, we share the enthusiasm for learning to learn with DFA, and hope our paper can motivate research in this direction.

**Reviewer #4**    The reviewer asks for details on the computational advantage offered by DFA. Assuming layers of similar sizes and sufficient parallel processing power, the speed-up factor over BP in the backward pass is equal to the number of layers, thanks to to to the parallelization of the backward pass. Furthermore, DFA reduces communication overhead where the model is spread across multiple devices. Advantages of this "backward unlocking" are laid out in Jaderberg, et al. 2016. We can't reduce this to an overall factor, as many specifics of the architecture come into play.

The reviewer inquires about the adaptation of DFA to architectures beyond the vanilla MLP used in Section 2. The strategy used for attention layers is presented in Appendix D, and adaptations to the random matrix for graphs and NLP are described in the Appendix C (lines 639 and 654). These will be added to the main text in the camera-ready version. More broadly, we introduce a random feedback $\mathbf{B}e$ after every non-linearity, in the spirit of Nøkland, 2016. We do not introduce any specific structure or operation to build the feedback. The method is applicable to any computational graph. We use a unique global feedback matrix (as in Launay et al., 2019), initialized from $\mathcal{U}(-1, 1)$, and normalized with the square root of the output dimension of every layer. We will add this detail in the camera-ready version. Regarding the necessity of the derivative of the activation, Gilmer et al., 2017 point to its importance in the update rule to perform better than a linear model. Concerning different activations functions, we stayed as close to the original models as possible (using ReLU), but Nøkland, 2016 shows that the method works for different choices of activation functions too.

[Meta-Review · NeurIPS 2020]

Strong empirical evaluation for Direct Feedback Alignment (DFA) on a broad range of tasks. The contributions are well summarized by R2's comments, "The work is motivated by arguing that DFA was so far only used on small datasets, and was shown to not perform well on computer vision tasks, in part because of the usage of CNNs in these settings. This survey challenges these views by conducting an extensive set of experiments using DFA to train s.o.t.a. models on s.o.t.a. benchmarks. The benchmarks include view synthesis, language modeling, recommender systems, and graph embedding. They compare the performance of these models to ones trained using a normal BP approach. The authors show that DFA can be competitive to classical BP in many scenarios, and also show how further improvements could be implemented. They also highlight potential benefits (e.g. parallelization) of training models with DFA vs BP." I agree with this. Authors have also addressed most of the reviewer's concerns in their rebuttal.